# A Survey on Compositional Learning of AI Models: Theoretical and Experimental Practices

**Sania Sinha**                                                              *sinhasa3@msu.edu*
*Department of Computer Science and Engineering*
*Michigan State University*

**Tanawan Premsri**                                                          *premsrit@msu.edu*
*Department of Computer Science and Engineering*
*Michigan State University*

**Parisa Kordjamshidi**                                                      *kordjams@msu.edu*
*Department of Computer Science and Engineering*
*Michigan State University*

**Reviewed on OpenReview:** *https://openreview.net/forum?id=BXDxwItNqQ*

## Abstract

Compositional learning, mastering the ability to combine basic concepts and construct more intricate ones, is crucial for human cognition, especially in human language comprehension and visual perception. This notion is tightly connected to generalization over unobserved situations. Despite its integral role in intelligence, there is a lack of systematic theoretical and experimental research methodologies, making it difficult to analyze the compositional learning abilities of computational models. In this paper, we survey the literature on compositional learning of AI models and the connections made to cognitive studies. We identify abstract concepts of compositionality in cognitive and linguistic studies and connect these to the computational challenges faced by language and vision models in compositional reasoning. We overview the formal definitions, tasks, evaluation benchmarks, various computational models, and theoretical findings. Our primary focus is on linguistic benchmarks and combining language and vision, though there is a large amount of research on compositional concept learning in the computer vision community alone. We cover modern studies on large language models to provide a deeper understanding of the cutting-edge compositional capabilities exhibited by state-of-the-art AI models and pinpoint important directions for future research.

## 1 Introduction

The compositional learning and reasoning of an intelligent agent refers to the ability to understand and manipulate complex structures by decomposing them into simpler parts and composing parts to form new complex concepts with a coherent understanding. This ability is a key factor in generalizing learning to unobserved situations (Hupkes et al., 2023). Compositional learning in intelligent systems is cognitively motivated since humans learn compositionally (Lake et al., 2019). Researchers have examined this phenomenon from cognitive, linguistic, and psychological perspectives (Shepard, 1987; Frankland & Greene, 2020).

The formal notion of compositionality originated from natural language and semantics, with various theories and arguments that elaborate on this concept. The principle of compositionality (Partee, 2004; Janssen & Partee, 1997; Montague, 1974) is defined as *"The meaning of a whole is a function of the meanings of the parts and of the way they are syntactically combined"* with three general methods- new meanings, new basic parts, and new constructions. One of the earliest formalizations of compositionality was grounded in grammar trees when cognitive scientists proposed an *information processing* approach to create a model

of the mind (Thagard, 2023). The birth of modern cognitive science happened following the proposal of phrase structure and transformational grammar (Chomsky, 1965). Compositional understanding of linguistic constructs has multiple aspects (Marelli et al., 2014; Li et al., 2021). For example, a nesting description such as "The black tall woman on the left of the car" conveys the intersection of multiple adjectives and spatial relations. Thus, the composition is defined as the intersection of multiple concepts. However, there are cases in which the direct intersection is not applicable, and the meaning should be inferred from the concepts in the global context, such as recognizing the sentiment of the following sentence "The pizza is so good, I hate this place!" Despite the natural language being a prominent manifestation of compositionality, this can be expanded to other areas of human intelligence, such as vision (Saffran et al., 2007). The same notion of intersection, as well as part-whole compositions, is essential for visual intelligence.

Compositional learning is important in complex tasks where high-level goals must be decomposed into smaller subgoals and plans, for example, when instructing an agent to navigate from one point to another (Schmidhuber, 1990). From the computational modeling perspective, traditionally formal grammars have been the means to address the compositional understanding of various modalities primarily in language and extended to vision (Girshick et al., 2011; Hong et al., 2021). While symbolic models inherently address compositional structures, using them alone, that is, parsing raw and noisy data into a structure with manually designed grammars will be brittle in real-world situations. Our study focuses on data-driven approaches based on artificial neural networks and the combined paradigms of neurosymbolic techniques (e.g. see Rajaby Faghihi et al. (2021); Premsri & Kordjamshidi (2024)). Several studies provide both experimental and theoretical analyses, indicating the competitiveness of the neural models in expressing compositional structures such as context-free grammars (Siegelmann & Sontag, 1995; Shi et al., 2021). Their expressive power added to the robustness in dealing with noisy data makes the neural techniques applicable to realistic situations.

In this paper, we examine multiple aspects of compositional learning, including compositional learning facets, datasets, computational models, and evaluation paradigms in both theory and practice. Figure 1 shows the scope and structure of our survey, covering four main topics, that is, Compositional Learning Facets, Datasets, Compositional Learning Models, and Evaluation. **For Compositional Learning Facets**, we overview the different measures of compositionality rooted in cognitive science that define abstract tasks for compositionality and connect them to the other adjacent topics such as continual learning and emergent intelligence. **For Datasets**, we cover existing benchmarks proposed in the AI community. They help bridge the gap between interdisciplinary theoretical definitions and the design of better evaluation benchmarks to pinpoint model capabilities. **For Compositional Learning Models**, we overview the architectures that aim to address compositional learning, divided into categories of basic neural architectures, large language models, and customized architectures, including neurosymbolic models. These models are mostly evaluated empirically using conventional benchmarks, while fewer studies conduct theoretical analyses. We cover both types of evaluations of various models when available in the existing literature. **For Evaluation**, we overview two evaluation approaches- theoretical and empirical. The theoretical evaluations examine various computational models in a mathematical framework, investigating their expressivity, and capacity for compositional learning, and analyze the generalizations to unobserved situations by computing the error bounds. Empirical evaluations include experimental results on benchmarks set by datasets and tasks created primarily to highlight the core challenges of compositional learning for language and vision understanding. Such results often report performance measures on the tasks designed to test the cognitive aspects of compositionality.

Overall, the cognitive aspects lay the foundation of the concept of compositionality and define the different abstractions associated with it and the tasks that are designed accordingly. The empirical evaluations use these tasks to evaluate compositionality using experimental performance. However, some studies examine the mathematical analysis and functional form of the models independent from the datasets. Finally, both these evaluations are used to develop models that are capable of compositional learning.

## 2 Compositional Learning Facets

Compositionality is one important aspect of generalization as a whole (Hupkes et al., 2023). Cognitive Science and Linguistics literature have identified broad categorizations of tasks that define compositionality and can be used to evaluate the compositional reasoning of models. The foundations of human natural language lie

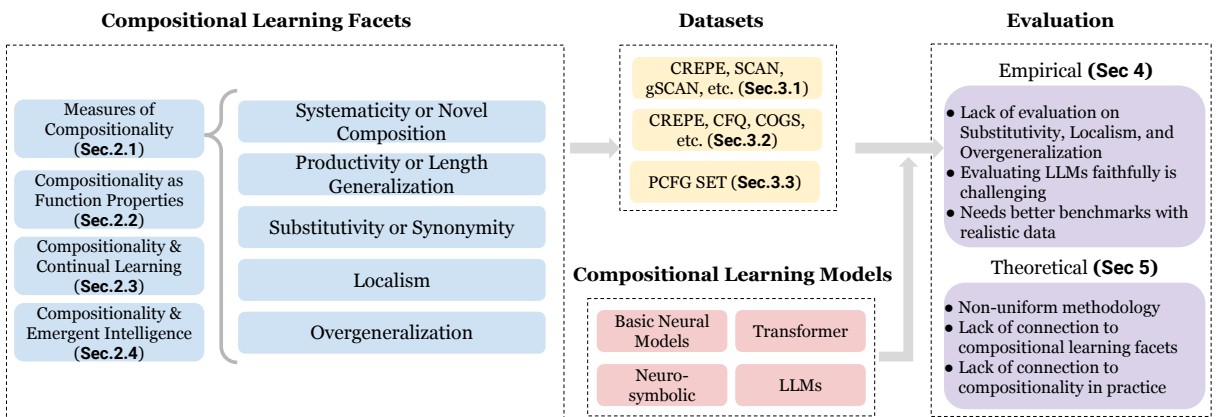

Figure 1: Outline of covered concepts in this survey, related to the structure of the paper. We structure our study of compositional learning by dividing it into four parts of compositional learning facets, datasets, compositional learning models, and evaluation methods from both empirical and theoretical perspectives. The topics have the respective sections associated with them. The main areas of required research and future direction are included in the descriptions in the Evaluation boxes, which are further discussed in Section 6.

in compositionality. A commonly used task categorization, derived from reformulated theoretically grounded tests from Hupkes et al. (2020), defines five main metrics of compositionality: systematicity, productivity, substitutivity, localism, and overgeneralization (Dankers et al., 2021).

## 2.1 Measures of Compositionality

Hupkes et al. (2020) introduces theoretically grounded tests for compositionality of models based on different interpretations of compositionality (Fodor & Pylyshyn, 1988; Chomsky, 1956). Some of these tasks existed in earlier research under different terms, such as productivity, which builds on a substantial body of prior research on length generalization. These tasks are becoming widely accepted tasks for compositional learning, and there are current datasets that use them for their evaluation splits. We describe these measures in detail below.

### Systematicity or Novel Composition

Systematicity is one of the most commonly used notions of compositionality in evaluating the performance of computational models. It has been defined as the ability to systematically recombine known parts and rules (Dankers et al., 2021). It derives directly from the commonly accepted definition of composition, which is the formation of compound expressions as a function of simpler ones (Partee, 2004). Systematicity is a standard concept in cognitive science research on building cognitive architectures that try to formalize the human thinking process (Fodor & Pylyshyn, 1988). The ability to syntactically combine known elements to form new or "unseen" expressions is an integral test when evaluating a model's ability to reason compositionally. It is also called Distribution-Based Compositionality Assessment (Keysers et al., 2020), where two principles are defined: one is to make sure the distribution of atoms is similar in both training and test sets, while another is to ensure the distribution of compounds is different. For example, if "red" and "car" are two separately learned concepts, the model should be able to accurately utilize the unseen concept of "red car."

### Productivity or Length Generalization

Another commonly explored test for compositionality is length generalization or productivity, as defined in Hupkes et al. (2020). In this evaluation, models are tested on their performance with expressions or sequences that are longer than training data. Longer input sequences may be recursive or nested versions

of seen phrases in the case of natural language inputs (Kim & Linzen, 2020). For example, if the model has seen "A and B," it should be able to understand "A and B and C."

### Substitutivity or Synonymity

Substitutivity is another form of evaluation defined in Dankers et al. (2021), which concerns the model performance on the introduction of synonyms in expressions. For example, testing a model on the translation of the same sentence, when switching between synonymous words such as donut to doughnut or aubergine to eggplant the translation would not change. This is one of the less explored axioms of compositionality.

### Localism

Another nuance of compositionality is the notion of global versus local composition. According to the principle of compositionality, the locality of the composition operator can vary. The meaning of a complex expression can depend solely on the meaning of its immediate parts (local composition) or the global structure of the context. Localism can be tested by analyzing the meaning a model assigns to a standalone compound versus when that compound is part of a larger expression. For example, sentences X and Y have the same truth values, but if the same context is added, their composition with the new context might lead to different truth values. For instance, when we add the context, "Peter thinks" and obtain "Peter thinks X" and "Peter thinks Y," these two new sentences can have different truth values (Hupkes et al., 2020; Carnap, 1947). The local interpretation of compositionality says that these new phrases will have the same truth values which might not be the case anymore as Peter might be aware of X and not Y. In other words, considering the phrase that X and Y are a part of, changes the meaning. This is another one of the less explored axioms of compositionality.

### Overgeneralization

Overgeneralization, as defined in Dankers et al. (2021), evaluates how much a model prefers an exception versus a rule. The term is originally used in language acquisition literature, also known as overregularization. A well-known example of this is the past-tense debate (Marcus et al., 1992) in language, which is about the rule that English past-tense verbs can be formed by appending -ed to the stem of the verb in most cases while there are some exceptions. This property can be evaluated by testing a model on exceptions of a usual rule in the training data and seeing if the model has over-fitted the training samples (Hupkes et al., 2020). Another example of this task is translating idioms where the meaning of sentences is "exceptions" to usual rules. For example, when translating the idiom "it's raining cats and dogs", a literal translation does not make sense as the phrase is an exception and has a specific meaning different from the usual literal interpretation. In this scenario, a model can achieve better performance by considering sentences in a global sense -that is looking at the bigger picture, such as context from placement in a compound- instead of trying to evaluate the meaning locally, such as by isolating the phrase. This is yet another less evaluated axiom of compositionality.

## 2.2 Compositionality as Function Properties

In Ram et al. (2024), compositional functions are defined with several components with a *computation directed acyclic graph* (**cDAG**) at the core. Their formal definitions facilitate the evaluation of compositional properties of the learning models (i.e. compositional functions). They relate their defined structure to the learning models' expressivity and sample complexity. In this system, **Systematicity** can be thought of as the expressivity of a compositional function as a low entropy program (for example, decision tree) versus a high entropy program (for example, transformer). **Productivity**, in simple terms, can be interpreted as whether a compositional function is recursive. **Substitutivity** tests whether a compositional function respects important abstractions and can be factored over them. **Localism** measures the stability of a compositional function against local changes, where the structure of the function's elements affects the importance of the level of locality respected. **Overgeneralization** is the extent of compression of a function, where a function might have a general rule but have exceptions to those in special cases.

### 2.3 Compositionality and Continual Learning

Compositionality is an important aspect of continual learning, also known as lifelong learning (Mendez & Eaton, 2021). Continual learning (Wang et al., 2024) is the concept of learning new tasks while retaining knowledge from previously learned tasks. In continual learning, compositionality is particularly crucial to prevent catastrophic forgetting, where earlier tasks are forgotten over time due to learning of new tasks (Liao et al., 2023). Since knowledge about novel tasks is compositional, the existing information can be combined in novel ways and used for future tasks. This enables forward transfer of knowledge rather than catastrophic forgetting (Mendez & Eaton, 2023). Both compositional and continual learning share the key challenge of finding reusable knowledge and further connecting those for transfer learning and dealing with complex unobserved situations.

### 2.4 Compositionality and Emergent Intelligence

The term emergent behavior has been used across various science-related fields, rooted in "More Is Different" by Nobel Prize-winning physicist P.W. Anderson (Anderson, 1972). Its introduction to the language modeling community, specifically in the context of the large language models, begins with Wei et al. (2022). They defined emergent ability as the ability that appears only in large models and is not observed in any smaller models. The emergent abilities demonstrated in their study include performing unseen tasks by following instructions (Ouyang et al., 2024) and demonstrating multi-hop reasoning skills through Chain-of-Thought prompting (Wei et al., 2024). These abilities reflect the model's capacity for generalizing to unobserved situations, which can further extend to the model's compositional learning ability. Therefore, the compositional learning ability of the models can be associated with the emergent intelligence of language models. In the same paper of Wei et al. (2022), creating new compositional learning benchmarks is proposed as one direction for evaluating and understanding LLMs' emergent abilities.

## 3 Abstract Tasks and Datasets

In this section, we categorize the existing datasets proposed for the evaluation of compositional learning. Our categorization is based on the type of compositionality facet explained in Section 2. Table 1 points to a list of important datasets we have surveyed. In general, there are more common evaluation benchmarks for Systematicity and Productivity. Systematicity focuses on the novel composition of seen atomic concepts and there are several benchmarks established for its evaluation, although some of those works do not explicitly use the term systematicity. The productivity measure was often referred to as length generalization before the term became commonly used. However, despite the abundance of datasets for these tasks, most rely on synthetic data, which poses a risk to their reliability in capturing the variation and complexity of real-world problems. While some datasets in the computer vision community for compositional learning utilize realistic images, they address fewer aspects of compositionality (e.g. object-attribute combination) compared to synthesized linguistic corpora designed for the same purpose. We describe some of the existing datasets and tasks in this section.

### 3.1 Systematicity or Novel Composition

**CREPE.** This is a Compositional REPresentation Evaluation benchmark (CREPE) (Ma et al., 2022). The dataset is synthesized and includes multiple splits, one of which relates to Systematicity. The main task setting is that, given an image, the model needs to identify an appropriate text caption describing it among multiple given choices. This systematicity challenge tests whether the model can systematically generate new combinations of seen atomic concepts during training. For example, "Crepe on a skillet" is never observed in the training while Crepe and skillet are observed separately in different contexts.

**SCAN.** The task is to navigate in a two-dimensional grid world based on natural language instructions. It is the *Simplified* version of the *CommAI Navigation* tasks (SCAN) (Lake & Baroni, 2017; Mikolov et al., 2018). One of the proposed experiments in SCAN evaluates the model's compositional generalization across primitive commands. Specific compounds are excluded from training where the model has seen the primitives and similar compound structures. Then, these unseen compounds are tested during testing.

**gSCAN.** The task is to navigate in a two-dimensional grid world based on natural language instructions (Ruis et al., 2020), which is a grounded version of SCAN (gSCAN). It includes 9 splits A-I. Categories B to H present tasks that form tests for systematicity (B,C- Novel Composition of Object Properties, D- Novel Direction, E- Novel Contextual References, F- Novel composition of actions and arguments, G,H- Novel Adverbs). Each split focuses on some form of novel composition of known concepts.

**PCFG SET.** PCFG SET stands for Probabilistic Context Free Grammar String Edit Task. It is an artificial translation task where sequences produced by probabilistic context-free grammar need to be translated into sequences representing their meaning (Hupkes et al., 2020). The output sequences can be constructed recursively using specified string edit operations applied to the input sequence, e.g., the input "repeat ABC" will be mapped to the sequence "ABCABC." The systematicity test uses a combination of words $a$ and $b$ in the input where the model has seen $a$ but not $b$ and vice versa. However, the combination of $a$ and $b$ is plausible in the corpus.

**COGS.** This dataset is designed for the Compositional Generalization Challenge (COGS) (Kim & Linzen, 2020). The task is a kind of semantic parsing based on a fragment of English where the models need to determine a formal meaning representation of the input English text. There are 4 categories in COGS that test some form of systematicity, including Novel Combination of Familiar Primitives and Grammatical Roles, Novel Combination of Modified Phrases and Grammatical Roles, Verb Argument Structure Alteration, and Verb Class.

**ReaSCAN.** This extension of gSCAN overcomes some of its limitations by requiring compositional language interpretation and reasoning about entities and relations (Wu et al., 2021). The challenges for systematicity are Category A, which tests novel object attribute combinations such as novel color modifier, color attribute, and size attribute, which is adapted from gSCAN (Ruis et al., 2020). Category B tests unseen co-occurrences of objects and relations, which is unique to ReaSCAN.

**SQOOP.** This is a dataset of Spatial Queries On Object Pairs (SQOOP). It is a minimalistic visual question answering, with yes-no answers. The input is an image and a question based on spatial reasoning (Bahdanau et al., 2019). Models are tested for answering questions on all possible object pairs after being trained on only a subset. Questions are of the form X R Y (X and Y are objects while R is a relation) where training sets are generated by controlled sampling of X and Y objects.

**CLUTRR.** This dataset is on Compositional Language Understanding and Text-based Relational Reasoning (CLUTRR). The task is, given a natural language short story, to answer questions on kinship relations that can be inferred from story (Sinha et al., 2019). Models are tested on combinations of held-out reasoning rules that are unseen during training. Thus, it tests systematic generalization capability or systematicity.

**KiloGram.** The task is a reference game task where given a textual description, the model has to select the appropriate image from a set of images. These images are tangrams, and the dataset has rich annotations of these images in two splits, FULL and DENSE, which have varying numbers of annotations, hence the name Kilo Tangram KiloGram) (Ji et al., 2022). There are different variations of this task such as showing parts of the image versus the whole image and making it grayscale versus colored. This dataset is an example of compositionality in vision since the whole tangram image is made out of parts, and the model learns how different parts combine to form different images.

**CompMCTG.** To evaluate the compositional learning of generative language models, Compositional multi-aspect controllable text generation (CompMCTG) benchmark(Zhong et al., 2024) is proposed. The task is to generate sentences given a set of concepts, including aspects of sentiment, topic, tense, person, and stuff. The benchmark tests systematicity by evaluating the model's capability to generate sentences with novel or unseen combinations of such concepts. For example, if the model has seen sentences with concepts of (red, car) and (blue, hat), it should be able to generate sentences for (blue, car). To perform these evaluations, they split the dataset into in-distribution (I.D.) and compositional, which have no intersection, although recombining elements in the I.D. set can form elements in the compositional set. Their evaluation protocol includes three test splits of Hold-Out, Attribute Compound Divergence (ACD), and Few-Shot evaluation. It is based on four popular textual corpora- Amazon Reviews (He & McAuley, 2016), a combination of IMDB, OpeNER, SenTube: Mixture (Liu et al., 2022a), YELP (Yelp, 2014), and Fyelp (Lample et al., 2019).

**MIT-States.** In this evaluation benchmark (Isola et al., 2015), the general task is to explain a collection of images in terms of the novel composition of primitive states and transformations applied to objects. It includes three different tasks: discovering relevant transformations (such as slicing, wilting), parsing states (such as sliced, wilted), and finding smooth transitions. It tests systematicity in image concepts such as being able to identify "sliced tomato" versus "whole tomato" and generalize it to unobserved compositions such as "sliced apple" versus "whole apple." There exists other similar works (Yu & Grauman, 2014; 2017; Tokmakov et al., 2019; Xu et al., 2023b; Bao et al., 2024; Xu et al., 2023a) in the vision community that evaluate such compositional concept learning.

**Skill-Mix.** There is no commonly used benchmark for the evaluation of compositional learning abilities of LLMs. A very recent benchmark, called Skill-Mix (Yu et al., 2024) covers some aspects of compositional evaluation for generative models. The task expects the models to generate text by combining various skills and imposing some constraints on the generated text. While the notion of compositionality is not highlighted in the paper, we categorize their tasks under systematicity.

## 3.2 Productivity or Length Generalization

**CREPE.** The productivity split of CREPE (Ma et al., 2022) evaluates if a model can perform the trained task of longer sets of expressions. In this task, there are variations of complexity for n-subgraphs with $n \in \{4, 5, .... 12\}$ and variations in the type of hard negatives used in the generation of text options. There are three types of hard negatives used- atomic hard negatives, swap hard negatives, and negation hard negatives.

**PCFG SET.** For the Productivity test (Hupkes et al., 2020), the data is split based on sequence lengths. The model is tested on sequences longer than the ones observed during training. For example, in a grammar context, if the model has been trained on the syntax entity-relation-entity, it will be tested on a longer, nested version of this concept, entity-relation-(entity-relation-entity).

**CFQ.** This is a dataset of Compositional Freebase Questions (CFQ). It is a natural language question-answering task (Keysers et al., 2020), focusing on semantic parsing, with a SPARQL query against the Freebase knowledge base. Questions are generated at varying levels of complexity. Various splits are available based on input length, output length, input pattern, or output pattern. All these splits aim to maximize compound divergence while minimizing atom divergence. This task appears to test both systematicity and productivity to some extent, although not explicitly. However, it cannot identify specific areas where a model's compositional behavior may be deficient.

**COGS.** While some splits of this dataset are explained in Section 3.1, it also contains a split for length generalization. Category 3 (i.e. Deeper Recursion), one of the splits of this dataset, (Kim & Linzen, 2020) is a test for length generalization by increasing the length of the input sequence recursively during testing. Input sequences are generated using nesting of phrases that are longer than those seen during training.

## 3.3 Other Generalization Criteria

To the best of our knowledge, PCFG SET (Hupkes et al., 2020) is the only benchmark that evaluates the other three additional criteria. The **Substitutivity** or synonymity test uses an input sequence with an atomic unit replaced by a synonymous atomic unit to evaluate how the model prediction changes. **Localism** is tested by using input sequences composed of smaller sequences A and B. The model is used to translate the full sequences first and then forced to process A and B separately. The outputs of these two experiments are compared to evaluate how local versus global the model is in its compositional reasoning. **Overgeneralization** test evaluates the model's results on input sequences that do not conform to the general rules of the dataset, that is, input sequences that are exceptions to the dataset rules. For example, the acquisition of past-tense forms such as the common "ed" ending (open-opened) versus more uncommon forms such as break to broke.

| Name | Text | MM. | Sys. | Prod. | Subst. | Loc. | Overgen. | References |
|---|---|---|---|---|---|---|---|---|
| PCFG SET | ✓ | ✗ | ✓ | ✓ | ✓ | ✓ | ✓ | Csordás et al. (2021) |
| CLUTRR | ✓ | ✗ | ✓ | ✗ | ✗ | ✗ | ✗ | Gontier et al. (2020), Minervini et al. (2020) Sinha et al. (2019) |
| SQOOP | ✓ | ✓ | ✓ | ✗ | ✗ | ✗ | ✗ | D'Amario et al. (2021), Bahdanau et al. (2019) |
| CFQ | ✓ | ✗ | ✗ | ✓ | ✗ | ✗ | ✗ | Furrer et al. (2021), Herzig et al. (2021), Liu et al. (2021), Cao et al. (2022) |
| SCAN | ✓ | ✗ | ✓ | ✓ | ✗ | ✗ | ✗ | Korrel et al. (2019), Nye et al. (2020), Dessì & Baroni (2019) |
| COGS | ✓ | ✗ | ✓ | ✓ | ✗ | ✗ | ✗ | Haurilet et al. (2019), Wu et al. (2023), Klinger et al. (2024) |
| gSCAN | ✓ | ✓ | ✓ | ✓ | ✗ | ✗ | ✗ | Gao et al. (2020), Spilsbury et al. (2024) |
| ReaSCAN | ✓ | ✓ | ✓ | ✓ | ✗ | ✗ | ✗ | Kamali & Kordjamshidi (2023) |
| CREPE | ✓ | ✓ | ✓ | ✓ | ✗ | ✗ | ✗ | Lin et al. (2024), Singh et al. (2023) |
| KiloGram | ✓ | ✓ | ✓ | ✗ | ✗ | ✗ | ✗ | Kojima et al. (2023), Gui et al. (2023) Ji et al. (2022) |
| CLEVR | ✓ | ✓ | ✓ | ✗ | ✗ | ✗ | ✗ | Bahdanau et al. (2020), Johnson et al. (2016), Niemeyer & Geiger (2021) |
| MIT-States | ✗ | ✓ | ✓ | ✗ | ✗ | ✗ | ✗ | Xu et al. (2023a), Naeem et al. (2021) |

Table 1: A summary of datasets and the compositional aspects they address with references of relevant papers on compositionality using these datasets. MM: multi-modal, sys: systematicity, prod: productivity, subst: substitutivity, loc: localism, overgen: overgeneralization

# 4 Empirical Findings: Compositional Learning Models

Traditional symbolic AI models naturally support compositional reasoning using classical logic applied to formal language (Szabó, 2022), formal verification (Giannakopoulou et al., 2018), and more. First-order logic can express objects, their properties, and complex compositional relations. Logical operations like conjunction, disjunction, and implication can express compositional structures on which inference rules are applied, supporting complex compositional reasoning (Porto, 2002). Another classical symbolic formalism includes grammars (Chomsky, 1965), which can express and generate complex compositional structures. However, dealing with noisy and uncertain data is hard with pure symbolic AI. Moreover, probabilistic augmentations and structured output prediction models have been able to explicitly model structural dependencies and support compositional reasoning based on their learned complex patterns from the data (Pearl, 1988). Nevertheless, scalability becomes a challenge for training and inference as the structural dependencies and the number of correlated variables increase. Given these long-lasting challenges of traditional models of compositionality, current neural models have shown success in both scalability and dealing with noisy and sensory data (OpenAI, 2024). Especially in modern large language models, complex compositional patterns can be memorized and resemble compositional reasoning. In the rest of this section, we overview the research focused on the development, design, and empirical evaluation of different types of neural models for compositional reasoning. We relate the type of empirical evaluations to the cognitive aspects of compositionality that they are testing. While some of these models utilize tasks and datasets covered in Section 3, others have their own tasks, specific to the problem of their choice.

## 4.1 Basic Neural Models

In Hupkes et al. (2020), different neural models are tested on a set of compositional learning tasks. They evaluate Long short-term memory (LSTM) Networks (Hochreiter & Schmidhuber, 1997), Convolutional Neural Networks (CNN) (LeCun et al., 1989), and Transformers for sequence-to-sequence language processing tasks on their proposed PCFG SET tasks including systematicity, productivity, substitutivity, localism and overgeneralization. On average, Transformer outperformed the other two models, but within the two classic neural models, the convolutional model performs better than the LSTM counterpart. In the reported results, two specific architectures, called LSTMS2S and ConvS2S were used. LSTMS2S is a recurrent, bidirectional encoder-decoder model with attention where the encoder and decoder are LSTMs, from the OpenMT-py framework. ConvS2S is a convolution-based sequence-to-sequence model as used in Gehring et al. (2017).

Several other works (Hupkes et al., 2018; Zheng & Lapata, 2021; 2022; Lake & Baroni, 2017) have used similar models to conclude that neural sequence models can exploit recursive compositional structure (Bowman et al., 2015; Irsoy & Cardie, 2014) in solving tasks. The related work in compositionality in computer vision indicates similar results. In Klinger et al. (2020), MLP, CNN, ResNet (He et al., 2015), and relational networks such as WReN (Barrett et al., 2018) and PrediNet (Shanahan et al., 2020) are evaluated on PCFG SET substitutivity and productivity tests. Their results indicate that compositional reasoning is challenging for the evaluated models and calls for more sophisticated architectures.

## 4.2 Transformer-based Architectures

The compositional capability of large language models is currently a controversial topic. They have been evaluated on general tasks such as arithmetic, logic, and dynamic programming that are compositional by nature Nafar et al. (2024a;b). Some of these evaluation efforts conducted in Dziri et al. (2023) concluded that GPT (OpenAI, 2024) family models, solve these tasks by reducing them to linearized subgraph matching, without developing true compositional reasoning skills. Moreover, it is shown that, asymptotically, they have architectural limitations in solving highly complex compositional tasks with novel patterns due to error propagation of the composition of erroneous building block functions. There is a substantial gap in the performance of Transformers on in-domain and low-complexity compositional examples versus out-of-domain instances. The tested tasks were 1) multi-digit multiplication (Hiebert, 2013), 2) Einstein's puzzle, which is a constraint satisfaction problem (Prosser, 1993), and 3) NP-complete maximum weighted independent set problem (Kleinberg & Tardos, 2005). These tasks are mostly aimed at testing systematicity and productivity. Their results indicate that transformers make predictions on shallow reasoning and memorization of similar subgraph patterns seen during training as opposed to reasoning holistically based on true compositional reasoning. This hypothesis also aligns with the findings presented in Chang & Bisk (2024). They conduct experiments with counting problems, a basic form of length generalization tasks. While transformers can count in observed cases, they dramatically fail to perform out-of-domain for the same task, indicating that transformers rely on memorizing observations. The results on transformers often extend to transformer-based language models. As shown in this work (Anil et al., 2022), fine-tuned transformer-based large language models lack generalization capabilities irrespective of the model size, when tested on other length generalization tasks such as parity and boolean variable assignment. This is due to the transformer's tendency to learn non-sequential patterns that do not apply to longer sequences. However, combining a pretrained LLM's in-context learning ability with scratchpad prompting significantly improves performance on longer sequences. This also implies that despite having access to an infinite data pool, LLMs can potentially learn some tasks better through in-context learning than finetuning, confirmed by theoretical work on LLMs explained in Section 5.

While a series of research papers focused on evaluating the compositional generalization of Transformers (Dehghani et al., 2019; Hahn, 2020; Feng et al., 2023) and complex reasoning Mirzaee et al. (2021); Mirzaee & Kordjamshidi (2022), some recent research investigated specific architectural factors that can impact the performance of Transformers on compositional tasks, following the claim that Transformers cannot reason compositionally (Dziri et al., 2023). In Ontanon et al. (2022), five configurations were evaluated on several different datasets and benchmarks, by varying five properties of Transformers including, 1) type of positional encoding, 2) use of copy decoders, 3) model size, 4) weight sharing, and 5) use of intermediate representations for prediction. The employed tasks were Addition, AdditionNegatives, Reversing, Duplication, Cartesian, Intersection, SCAN-length and SCAN-add-jump, PCFG productivity and systematicity, COGS, and CFQ-mcd1. This work concluded that relative positional encodings usually help, but using embeddings is necessary, and merely relative position biases are not sufficient. Tasks like SCAN and CFQ were not affected by positional embeddings. Tasks like Duplication or PCFG benefit from a copy decoder because it can learn a type of symmetry like learning a certain position of the input. As for model size, it was found that for algorithmic tasks, large models did not make a difference. However, for PCFG, large models seemed to outperform their smaller variants. Weight sharing across transformer layers seems to improve accuracy in most tasks. Intermediate representations also improved performance by creating new levels of abstraction that make reasoning easier for solving the end task. Specifically, using intermediate representations achieved state-of-the-art performance on COGS by converting the task from seq2seq to sequence tagging. Using intermediate representation on CFQ, eliminating the need to perform Cartesian products

by using triple representations, also showed a significant performance improvement. However, intermediate representations need to be crafted specifically for a task and were only tested on COGS and CFQ. Other works (Ruoss et al., 2023; Kazemnejad et al., 2023; Chang & Bisk, 2024) also investigate the effect of different positional encodings on the performance of Transformers for specifically length generalization. Kazemnejad et al. (2023) concludes that commonly used positional encodings such as ALiBi and APE are not well suited for downstream tasks but transformers without any positional encoding (NoPE) outperform other explicit positional encoding methods when testing decoder-only transformers on tasks such as SCAN and copying and reversing. Despite this, Chang & Bisk (2024) presents compelling evidence suggesting that each type of positional encoding demonstrates robustness on unique length generalization tasks. Consequently, the performance of Transformers on compositional tasks varies depending on the selected positional encoding. The authors even suggest integrating multiple positional encodings into Transformers for better compositionality by combining different strengths of those encodings.

Another type of research deviates from evaluating current conventional architectures and instead focuses on designing a new architecture that can compositionally generalize better. One of such models, a multi-modal transformer called GroCoT (Grounded Compositional Transformer) (Sikarwar et al., 2022), was designed and achieved state-of-the-art results on GSRR and ReaSCAN. Another multi-modal transformer from Qiu et al. (2021) is used as a backbone model with changes to Encoder, Decoder, modified spatial representation, interleaving self-attention, and a modified world state encoding. This work also showed that a single-layer transformer with a single head can ground and compose novel combinations of visual object attributes. They tested the generalizability on RefEx, that is on grounding referring expressions and, their proposed evaluation benchmark based on the target identification subtask of ReaSCAN. Another customized architecture example is adding Pushdown Layers to transformer architecture (Murty et al., 2023) which was recently presented as a replacement for standard self-attention. The recursive structure of natural language is challenging for self-attention layers to capture due to the lack of an explicit recursive-state tracking mechanism, which Pushdown Layers try to overcome. Pushdown layers have a stack tape that helps them model the recursive state of language. This helps Transformer-based language models softly modulate attention over tokens when predicting new ones. Other architectures derive inspiration from cognitive sciences. In the same line of work, RegularGPT (Chi et al., 2023) takes inspiration from working memory. It modifies the Transformer architecture to use weight sharing, adaptive depth, and sliding-dilated attention for better length generalization. When tested on the task of natural language extrapolation, it was found that it captures the local windowed attention patterns, which previous work identified as essential for the task. Additionally, it can efficiently model regular languages such as PARITY.

### 4.3 Neurosymbolic Architectures

A rising trend in cutting-edge research on modeling intelligent systems is neurosymbolic modeling. As the need for general-purpose AI models grows, there is a need for highly compositional models that can reason based on previously trained simpler tasks to do novel and complex ones. Although not explicitly mentioned in this research, they mainly address systematicity and productivity. One approach in this direction is to use natural language explanations to generate formal specifications that explicitly lay out a compositional task in terms of required simpler steps. The formal specifications then are passed to appropriate *engine*s to solve the problem. A prominent vision understanding model that follows this approach is VisProg (Gupta & Kembhavi, 2023). VisProg is a modular neurosymbolic model that can solve various compositional visual reasoning tasks given natural language instruction relying merely on the in-context learning of large language models. It produces modular programs in Python to obtain the solution. This approach provides an interpretable reasoning for how the model derives the solution. These modular programs use built-in modules supported by VisProg such as off-the-shelf neural computer vision models, image preprocessing modules, or Python subroutines, and solve complex tasks without any task-specific training. Another example in this line of work is to generate a formal logical specification of the problem from natural language explanations and pass the logical form to a logical reasoner engine (Poesia et al., 2023). This work uses large language models such as GPT-3 or GPT-3.5 Turbo, for producing "guides" to solving complex compositional tasks by breaking those down into smaller steps based on a reasoning chain. Similar to this, many recent works focused on different prompting strategies that can be used to solve complex compositional tasks with a modular approach. Examples include Decomposed Prompting (Khot et al., 2023), which uses a modular approach

to decompose a complex task into simpler sub-tasks via prompting and pass on these sub-tasks to LLMs that are capable of solving them. This method allows for the optimization of a prompt for a specific sub-task, which can be further decomposed, or replaced with more effective prompts, trained models, or symbolic functions as necessary. A similar approach for mapping to probabilistic logical reasoning is proposed in Nafar et al. (2024b). Neurosymbolic modeling has also been used previously in generative models for concept learning (Hofer et al., 2021) such as in the context of auditory signals for learning evolved combinatorial structure in language.

Another branch of Neurosymbolic modeling explores leveraging the ability of large language models to do reasoning. This includes casual reasoning with an introduced benchmark, CLadder (Jin* et al., 2023). Their approach is to provide step-by-step structured prompts as a form of a chain-of-thought strategy called CausalCoT. The chain of thought (COT) conveys the formal symbolic representation of the causal reasoning problem. The CoT prompting is influenced by natural language rationales or reasoning processes (Qiao et al., 2023), which is similar to the use of answer rationales in inducing arithmetic problems (Ling et al., 2017).

**Compositional Neural Architectures.** Over the years, several architectures and theories have been proposed to model compositionality in their design. This aligns with the idea that neural architectures are structurally compositional (Lepori et al., 2023), meaning they leverage subroutines to break down complex tasks. One of the earlier examples of this type of architecture includes neural modular networks (Andreas et al., 2016). Neural modular networks were designed to model the inherent compositionality that exists in linguistic structures. The conceptual modules are built in the neural architecture based on the problem specification. For example, in a visual scene understanding problem, we can place modules for detecting objects, their compositional properties, and relations, which are the main building blocks for abstract reasoning needed for complex scene understanding.

In a similar line of work (Kuo et al., 2020), a network architecture was built that is compositional in nature and makes it possible to interpret what each part of the network learns. It solves tasks in gSCAN, which has agent navigation tasks in a 2D environment following a natural language command. The neural architecture built in this work assembles a *command-specific* network from previously trained modules, modeling the compositional nature of the command (task). Later research showed that in Neural Module Networks, it is hard to make the designed modules faithful to expressing the concepts that they are designed for. This is despite the overall network achieving high accuracy for the target task (Subramanian et al., 2020).

Given the importance of compositional learning in lifelong and continual learning, neurosymbolic approaches have been a suitable framework to address continuality through combining program synthesis and neural modeling. One such example is HOUDINI (Valkov et al., 2018), which is a neurosymbolic framework for lifelong learning of tasks combining perception and procedural reasoning such as counting, summing, and shortest-path computation. They use program synthesis to search over networks described as typed functional programs for the given task, whose parameters are then tuned end-to-end based on stochastic gradient descent. Another example is the Logic-Enhanced Foundation Model (LEFT) (Hsu et al., 2023), a framework designed to learn grounding and reasoning across domains using a differentiable, domain-independent, first-order logic-based program executor. It addresses the lack of generalizability across domains seen in related work, such as VisProg, which we previously discussed, including limitations in generalizing concepts from 2D to 3D images.

**Neurocompositional Computing.** Neurosymbolic modeling has been motivated by its connection to *neurocompositional computing*. The term "neurocompositional computing" was coined in Smolensky et al. (2022b). It defines a type of computing that underlies human cognition as argued in contemporary cognitive science theories in Smolensky & Legendre (2006) and incorporates principles of Continuity and Compositionality. The Continuity Principle states that the encoding and processing of information should be continuous, that is, represented by real numbers that vary continuously and can be changed by arbitrarily small amounts. The Compositionality Principle states that larger, more complex structures can be decoded on the basis of smaller, simpler, and familiar building blocks. According to the *Central Paradox of Cognition*, the human brain follows both a continuous neural computing structure and a discrete compositional-structure computer.

Following this theory, neurosymbolic models that are both continuous and discrete in architecture seem like the ideal approach to modeling compositional behavior in computational models (Smolensky et al., 2022a).

# 5 Theoretical Findings: Mathematical Formulations of Compositionality

Theoretical analysis is fundamental for deepening our understanding of the compositionality of learning models. It can reveal intriguing and previously uncovered information that experimental analysis may overlook. Many research works have proposed diverse approaches for investigating the compositionality of learning models. We highlight three different approaches, including a mathematical framework for defining compositionality (Ram et al., 2023), exploring the upper-bounds of expressivity that relate to compositionality (Merrill & Sabharwal, 2023), and analyzing error-bounds to demonstrate the model's limitations in solving compositional learning problems (Dziri et al., 2023). In the rest of this section, we provide a detailed overview of these cases and explain the theoretical results on compositional generalization of classical neural networks, transformers, and modern language models. We also relate the mentioned techniques to aspects of compositionality when applicable.

## 5.1 Classical Neural Network

Ram et al. (2023) provides a mathematical definition of compositionality for learning models and connects their expressively to computational complexity. They frame the existing well-known models, such as variations of RNN and CovNets, with the provided formal definition to explain properties related to their compositional generalization. Hewitt et al. (2020) further investigates the RNN's ability to generate natural language with a certain nesting depth. They claim that RNNs with optimal memory and $O(m \log k)$ hidden units can generate a natural language of well-nested brackets of $k$ types and $m$ bounded nesting depth. With the rise of LLMs, compositional generalization has recently become more critical. Due to their large-scale parameters and training data, LLMs perform empirically well on many tasks. However, the empirical performance measures are now less reliable, as the high performance on test data can not be interpreted as compositional generalization anymore. This issue is due to the nature of internet-scale training of LLMs and data contamination. Consequently, there is more urgency for theoretical studies to understand their limitations and measure their reliability in unobserved situations. However, Ahn et al. (2023) argued that studying the smaller models at the single neuron level potentially leads to a better understanding of the large/deep models' learning behavior, which is related to explaining the Systematicity of the model. They also establish a connection between the *Edge of Stability* identified by the learning rate of the gradient descent approach for non-convex optimization and the emergent abilities in learning. This result remains limited to the scope of a single neuron and has not yet been extended to large models.

## 5.2 Transformers

To define the limitations of LLMs, it is essential to investigate the limitations of transformers and their underlying architectural component. In this work (Merrill & Sabharwal, 2023), the authors assume a specific transformer type, suggesting that their arithmetic precision is logarithmic in the number of input tokens. Based on this assumption, they demonstrate that transformers cannot accurately solve linear equalities or check membership in an arbitrary context-free grammar with empty productions. The studies of transformer precision have been explored before in Dehghani et al. (2019). They claim that standard transformers have limited precision, implying that they cannot handle an infinite input length. This conclusion indicates the limitations in of the compositionally of the transformers in terms of the Productivity aspect. Another notable theoretical investigations focus on the activation functions to explain the limitation of the transformer (Dehghani et al., 2019; Hahn, 2020). Hahn (2020) analyzes both hard-attention and soft-attention transformers. For hard attention, they prove that the transformer ignores most of the input information diagnosed by the specific modifications applied to the input. According to their analysis, transformers with hard attention will be unable to solve problems that require processing the entire input, such as PARITY and logical formula problems. However, this conclusion contradicts older papers that state transformers are Turing complete (Pérez et al., 2021). They utilize the strong assumption that all input information is accessible using hard attention to prove Turing completeness. This leads to a different conclusion, stating that the

transformer can compute and access the entire internal dense representation. Hahn (2020) also investigate the model's behavior with soft attention. They illustrate that Transformers struggle with solving long input by demonstrating the influence of input on output substantially drops as the input gets longer. This is a similar conclusion, using a different approach analyzed in an older study of (Dehghani et al., 2019). Based on these analyses, they further confirm the lack of productivity aspect of transformer architecture caused by limited training data lengths.

Despite proving weaknesses of the transformer, Hahn (2020) claims that the transformer has the potential to solve small input tasks completely. Two recent works also support this claim. The first work provides proof utilizing computation graphs and a theoretical study of error propagation in transformers. They claim that the auto-regressive transformer's error reduces as the size of the input decreases (Dziri et al., 2023). Moreover, they show that transformers reduce problems into multi-step compositional problems to solve larger tasks, which is strongly related to the Novel Composition of the compositional aspects. The second work supports the mentioned claim based on the study of sub-sequential finite-state transducers (SFSTs) (Valvoda et al., 2022). They generate a set of random SFSTs following Montague's Compositionality theorem to discover the coverage limitation. This limitation is inversely related to the size of the dataset and significantly impacts the probability of a model's successful generalization.

### 5.3 Large Language Models

In addition to inconclusive theoretical studies on transformer limitations, there are controversial results on large language models. The most noteworthy study is on the emerging abilities and capabilities claimed to be unique in the large models. The emerging abilities relate to the generalization to new and complex tasks in LLMs. This kind of ability is also a feature of models' compositional learning ability, allowing them to perform in novel compositional situations (Yu et al., 2024). Multiple works have shown the existence of emergent abilities of LLMs (Wei et al., 2022). The recent work of Arora & Goyal (2023) provides a mathematical framework for identifying complex skills in language models. They use the LLM Scaling Rule to argue that emergent skills are the results of reducing excessive loss. This excessive loss enables the model to learn how to utilize and combine skills from downstream tasks during training. Their claims are based on the assumption that language inherently contains a random mix of complex skills. Although several experiments reveal these emerging capabilities, at least two papers disclaim their existence. The first group provides a theoretical proof based on a mathematical framework. They illustrate that the emerging ability appears due to the selected evaluation metrics that are nonlinear and discontinuous (Schaeffer et al., 2023). They show as an artifact of the evaluation metrics, even simple models such as CNNs can show emerging abilities. Therefore, they conclude that emerging abilities may not be a fundamental property of the large models. Moreover, Lu et al. (2024) provides an extensive empirical study with 1000 experiments on 22 tasks with different LLMs. However, given the inconsistency in some results and the unpredictability of emerging abilities, they do not find any strong evidence of how they emerge. They associate the performance with in-context learning techniques, memorization, and data contamination. However, a recent study presents a positive theoretical analysis of reasoning capabilities by investigating the chain of thought (CoT) (Wei et al., 2024) and draws a different conclusion. They argue that the log-precision transformer can perform fundamental operations such as multiplication and a look-up table. Consequently, it can solve linear equations and other reasoning problems if it stores all the input information. However, the architecture alone struggles with storing the entire input, as observed in Dehghani et al. (2019); Merrill & Sabharwal (2023). They show that the model addresses this limitation by repeatedly referring to the input by enabling CoT (Feng et al., 2023). Therefore, with the right number of CoT examples, LLMs can overcome the transformer's weakness in solving mathematical reasoning. This is aligned with previous empirical results of in-context learning discussed in Section 4.2.

## 6 Discussion and Future Direction

There has been a large amount of research on the compositional learning ability of humans from a cognitive perspective (Fodor & Pylyshyn, 1988; Ito et al., 2022). Researchers in linguists and formal languages have formalized the notion of compositionality since languages have inherent compositional structure (Chomsky,

| | Model Type | Theoretical Analysis | Empirical |
|---|---|---|---|
| Basic Neural Models | RNN | Hewitt et al. 2020 | Bowman et al. 2015 |
| | CNN | ✗ | Hupkes et al. 2020 |
| | LSTM | Siegelmann & Sontag 1995 | |
| Transformer-based | (Customized) Transformers | Hahn 2020, Pérez et al. 2019, Dehghani et al. 2019 | Ontanon et al. 2022 |
| | LLM | Dziri et al. 2023 | Schaeffer et al. 2023 |
| Neurosymbolic | Neural Modular Network | ✗ | Kuo et al. 2020 |
| | Other Models | ✗ | Gupta & Kembhavi 2023 |

Table 2: Summary of computational models with compositional learning ability from the theoretical perspective and an example from the experimental perspective.

1965; 2002). However, from the AI and machine learning perspective, ideas are borrowed from both cognitive and linguistics, and computational tasks and models are designed focusing on narrow aspects of compositionality (Hupkes et al., 2020). Our investigation of AI models indicates several challenges regarding the designed tasks, benchmarks, and theoretical frameworks that make the evaluation of computational models problematic.

Figure 1 shows the connections between the main topics identified and discussed in this survey. Among the five identified types of compositional learning facets, only systematicity and productivity are well-researched and have clear connections to evaluation benchmarks. Given that these are the five main metrics of compositionality, we should aim to expand our evaluation capabilities by developing tests for the other three as well. Empirical evaluations are comparatively more well-studied compared to theoretical analyses. Theoretical evaluations are either lacking or do not follow a consistent methodology. There is a lack of connection between the theoretical methods and the cognitive aspects, making these results hard to use to guide better architectural design. For Models, different types of architectures have been designed. However, evaluating LLMs and making fundamental design decisions for compositional generalization present new challenges. We outline some of these challenges in detail below.

**Less Explored Facets of Compositionality.** Only Systematicity and Productivity have been well-researched and have established connections to evaluation benchmarks. While the other three were introduced as fundamental types of compositionality, they have received less attention, as they appear to be less commonly occurring aspects of compositionality. However, in the era of LLMs and the emergent in-context learning, Substitutivity and Localism are potential bottlenecks in the performance of LLMs for attending the appropriate context for solving problems. Moreover, Overgeneralization can be associated with hallucination and generating unfounded and incorrect information by making up new unrealistic abstractions. While hallucination is a broader concept than overgeneralization, this compositional learning facet can highlight an important aspect to be addressed to prevent hallucination (Huang et al., 2023). Therefore, directing attention to the other three types of measures can help establish new formal evaluation benchmarks, contributing to the development of more robust systems and addressing the challenges of the LLMs.

**Synthetic and Unrealistic Evaluations.** One issue in current evaluations is that controlled and clean tests of compositonality are mostly synthesized (Wu et al., 2021; Ruis et al., 2020). This is evidenced by our examples in Section 3. Even in rare cases that claim to work with realistic data (Keysers et al., 2020), synthesized questions are used to query knowledge graphs. However, more recent studies on language models' evaluation of compositionality focus on more challenging problems such as multi-hop question answering (Press et al., 2023; Liu et al., 2022b; Okawa et al., 2023; Mirzaee et al., 2021) as well as complex puzzles with combinatorial search solutions or compositional mathematical reasoning (Dziri et al., 2023). Although the benchmarks are designed for evaluating specific tasks, the reliance on mostly synthesized data risks the effectiveness of generalization to real-world data, which is often more complex.

**Misalignment of Performance and Compositional Learning (LLM Evaluation Challenge).** The second challenge that mostly applies to LLMs is data contamination. Though the recent research compares language models to the specialized architectures and indicates their outperformance in compositional tasks (Furrer et al., 2021), this result does not necessarily mean these models have better generalizations in

recognizing unobserved compositions (Press et al., 2023). A major issue with these evaluations on realistic data is the difficulty in disentangling the compositional reasoning from the data contamination (Sainz et al., 2023) and memorization. However, very little recent work has been done to address this issue (Yu et al., 2024). Based on our investigation of theoretical studies in Section 5, we identify that the generalization abilities can be an artifact of observing more complex data in larger contexts as well as evaluation metrics as has been pointed out in Schaeffer et al. (2023) and is a pertinent issue that needs to be addressed in new evaluation benchmarks.

**Inconsistent Theoretical Methodology.** Given the difficulty in obtaining conclusive empirical studies, theoretical understandings become even more important, nowadays. However, the lack of a well-established and practically informative theoretical framework for investigating the limitations and capabilities of LLMs has been a challenge to a deep understanding of their generalizability. According to our studies on the theoretical explanation of transformers in Section 5, their compositionality is still under discussion. Some results illustrate that transformers possess compositional generalizability based on their ability to solve complex tasks based on smaller subtasks (Feng et al., 2023; Dziri et al., 2023). However, other results based on a different evaluation methodology suggest that the emergence of such abilities, including compositional learning, is associated with the user's choice of evaluation metrics (Schaeffer et al., 2023). Similar lines of study deny the emergence of intelligence and relate the new abilities to in-context learning methods, models memory, and linguistic knowledge (Lu et al., 2024). Many research results confirm the limitations in the generalizability of LLMs. For example, the building blocks of these models, i.e. transformers, still have severe limitations in comprehending large inputs (Hahn, 2020; Dehghani et al., 2019). Despite these controversial discussions, there are only relatively few studies on the theoretical analysis of transformer-based models. The methodological frameworks for examining the LLMs' capabilities are not standardized. Therefore, the compositional capabilities of the current SOTA models require more attention from the research community. This research direction will help towards more consistent and conclusive results on the limitations of the model's generalizability, specifically the compositional generalization.

**Cognitive Motivation.** The fundamental capabilities of current AI models have been debated and criticized by scientists in cognitive science and psychology (Bender & Koller, 2020; Marcus, 2018). Despite giant leaps of performance progress in modern AI, there are distinct differences between these machines and human intelligence. From our exploration of cognitive aspects of compositional learning in Sections 1 and 2, we observe that cognitive foundations do not have strong ties to model building yet. Evaluating different models reveals that they often rely simply on pattern recognition (Geirhos et al., 2020; Dziri et al., 2023), instead of a holistic understanding of a problem grounded in reality and situation, as was seen in Section 5. Understanding human intelligence from Cognitive Science literature suggests that we must move beyond current engineering trends to build causal models of the world that support knowledge and understanding. The key ingredients of such human-like rich and efficient learning are compositionality and learning-to-learn (Lake et al., 2017).

## 7 Limitations

Despite the comprehensive nature of the survey and our efforts to cover and connect most research relating to compositional learning, we would like to acknowledge some limitations. The scope of this survey covers a broad spectrum of topics and tries to capture both theoretical and experimental frameworks, but there might be some relevant papers that are missed. Compositional learning is an interdisciplinary topic across Computer Science, Linguistics, Cognitive Science, etc. Although we have included insights and connections from across these fields, our work has a more in-depth focus on Computer Science literature, especially Natural language processing. While we tried to provide the overall picture of the related research and build a coherent story, we might not capture the detailed nuances of each definition and application.

## Acknowledgements

This project is partially supported by the Office of Naval Research (ONR) grant N00014-23-1-2417. Any opinions, findings, conclusions, or recommendations expressed in this material are those of the authors and

do not necessarily reflect the views of the Office of Naval Research. We thank Dr. Tim Klinger and the anonymous reviewers for their constructive feedback, which greatly helped us improve this manuscript.

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
