# OpenReview forum: "A Survey on Compositional Learning of AI Models: Theoretical and Experimental Practices"
_TMLR — Accepted by TMLR_

### Review · Reviewer_P9Jg · 2024-08-16

**Summary Of Contributions:**

This paper surveys compositional learning work from several aspects: cognitive aspects, task and datasets addressing cognitive aspects, compositional reasoning ability for various neural models, theoretical analysis centering model architectures, and some calls to address some challenges in compositional evaluation and theoretical analysis.

**Audience:**

Yes

**Claims And Evidence:**

Yes

**Requested Changes:**

See weakness above.

**Strengths And Weaknesses:**

Strengths:
1. It is a recent comprehensive compositional learning survey.
2. Considering the importance of studying compositionally in learning, and current hype of scaling law, it is a important positional survey to draw extra attention to this problem.

Weaknesses:
1. The paper collects work mostly from NLP perspective, despite some touch on multimodal compositional learning like gSCAN. The narrative from title, abstract and introduction appears to position this survey as compositional learning of AI model, so I think author should consider include more work from multimodal, and embodied ai perspectives. Compositionality in those domains needs more study in terms of metrics (cognitive aspects), evaluation, and model design.
2. For discussion, when model does not possess high compositionality but have high performance in terms of some evaluation, what could be the risk of this misalignment, author could consider survey works or provide discussion around this issue.

---

> ### Author Response · Authors · 2024-09-12
>
> Thank you for your valuable suggestions. We have taken those into account and worked on revising the paper. Overall, the revised parts of the paper are shown in blue text.
>
> **We have addressed the weaknesses in the following ways.**
>
> 1. **[including multimodal and vision]** We have added a line (in blue color) to our abstract to clarify our focus on NLP and then its combination with vision in the survey. We realized that natural language tasks cover a more variety of different compositional learning aspects compared to vision tasks. Vision tasks mostly focus on combining objects and attributes. We have added this justification to the paper in the introductory paragraph of Section 3 and also added a paragraph under Section 3.1, which references some of the most popular vision datasets for compositional concept learning and cites related work (MIT-states, etc.). This dataset has also been added to Table 1.
>
> 2. **[risk of misalignment]** We have modified the paragraph and renamed it to "Misalignment of Performance and Compositional Learning" to clarify this point more. We have also made a connection to emergent ability in Section 2.4. Additionally, we connect modern challenges such as hallucinations in "Less Explored Facets of Compositionality" under Section 6.

---

### Review · Reviewer_pMmS · 2024-08-18

**Summary Of Contributions:**

The paper discusses the cognitive aspects of compositional learning, how current evaluation protocols test for these aspects, and how current models perform under these evaluations.

**Audience:**

Yes

**Claims And Evidence:**

Yes

**Requested Changes:**

I am happy to change my Claims and Evidence evaluation if the claims are better scoped or substantiated. This paper will be much better if these claims are better fleshed out to be accurate and meaningfully relevant for practitioners.

Below are minor suggestions

1. When citations are not an object in the sentence (for example, the first four citations in this paper), ensure they are wrapped in parentheses. Based on your latex, this could be done using \citep instead of \citet or \cite. Other citations are not hyperlinked throughout the text.

2. The paper may want to discuss the recent empirical work of Skill-Mix (https://arxiv.org/abs/2310.17567) and corresponding theoretical work (https://arxiv.org/abs/2307.15936) that aim to quantify compositional learning for LLM's

**Strengths And Weaknesses:**

**Strengths**

1. The work defines which notions of compositionality it is studying. The sections are accordingly organized.
2. The work discusses a lot of prior research, datasets, and models concerning compositionality.

**Weaknesses**

1. Figure 1 highlights the claims of the paper. These claims contain most of the novelty in the paper, but each only gets one paragraph of explanation in the discussion. Some of these claims are strong and unsubstantiated, while others do not offer strong insight. In general, I think these claims are the most important part of the paper, and should offer deep insight accrued from knowing all the related topics.
    - The paper claims there should be more research should be done in benchmarking the latter 3 aspects of compositional learning. Though they show there is a lack of evaluation here, why is it important to benchmark these aspects? Sharing these reasons is important for future research.
    - The paper claims there should be better benchmarks. They specifically say existing benchmarks are synthetic/unrealistic, while continuing to show many benchmarks that address their issues. What are the issues with current benchmarks then?
    - There is a single paragraph in the discussion concerning inconsistent theoretical methodology. It claims that this inconsistency is because some say compositional learning exists, while others say it is unpredictable (sidenote, there are factual errors here. Schaeffer et al does not claim it is unpredictable). Why is this inconsistent.
    - Where is it justified that task-specific models have better expressivity?

2. Outside of these claims, the main contribution seems to be classifying existing works under the compositional framework proposed in Dankers et al 2021. I find survey papers most helpful when they can synthesize new viewpoints about the field and consider the connections between different works. Though this paper summarized existing contributions, I do not feel like I was left with a greater sense of understanding of the current state of compositional learning.

3. What is the connection between emergence and compositional learning? I did not follow the connection at the start of 5.3

TLDR: Though the work is relatively comprehensive, its main claims are given less attention, while the rest of the work consists of more summary than synthesis.

---

> ### Author Response · Authors · 2024-09-12
>
> Thank you for your feedback and for acknowledging the strengths. We have tried our best to detail our claims better and evidence them by connecting them to different sections of the paper. We have also included additional discussions to diversify the topics of compositionality that we discuss, such as Sections 2.3, 2.4, and "Less Explored Facets of Compositionality" in Section 6. Overall, the revised parts of the paper are shown in blue text.
>
> **We have addressed the weaknesses in the following ways.**
>
> 1. Thanks for the comment. To make the connection clearer, we have added new text in Section 6 in color blue that expands upon our claims and ties it back to our observations in other sections of the paper as evidence.
>     - **[latter 3 aspects]** The other facets of compositionality are tied to important aspects of model performance, especially for LLMs, and could potentially help mitigate risks such as hallucination and context understanding. We have added a new titled paragraph in color blue for "Less Explored Facets of Compositionality" in Section 6 to detail this claim.
>     - **[synthetic benchmarks]** The way the facets of compositionality are expressed in synthetic data might be much more limited compared to their realistic manifestation and pose risks for generalization. We have added new text in blue color to "Synthetic and Unrealistic Evaluations" in Section 6 to add more clarity to this. We have also added to the introductory paragraph of Section 3 to address this claim in the main body of the survey.
>     - **[inconsistent theoretical methodology]** Thank you for the comment. We tried to state that they disclaim the existence of emergent ability that is usually based on the large models' high accuracy. We modified the wording to reflect the accurate point of those two papers. Please see the proper term in the revised version in the blue color of "Inconsistent Theoretical Methodology" of Section 6.
>     - **[better expressivity]** This has been related to some internal discussions and needs an update in the figure. We do apologize for the confusion. The figure is modified, and task-specific is removed.
>
> 2. **[new viewpoints]** Figure 1 shows our own viewpoint, structuring the related literature from three main perspectives. Overall, we have covered a broad range of papers and made different connections in the survey with respect to the aspects that we have identified. However, the measures of compositionality are derived from Dankers et al. because, to our knowledge, it is becoming widely accepted and is one of the only formal measures of compositionality for computational learning models. We have clarified this point in Section 2, and thanks to the reviewers' feedback, we have added more facets under that section by explaining the connection between continual learning and emergency intelligence. We compiled a thorough discussion in section 6 and identified challenges based on our comprehensive literature review. In the new version, we have expanded Section 6 and connected our claims to the rest of the paper to substantiate them and highlight the evidence based on our reviewed literature.
>
> 3. **[emergence and continual learning]** To briefly reply, "The emerging abilities relate to the generalization to new and complex tasks in LLMs. This kind of ability is also a feature of models' compositional learning ability, allowing them to perform in novel compositional situations." In revising the paper, we added a new section (2.4) to clarify this connection, pointed back to this in section 5.3, and added more citations.
>
> For the requested changes **[TLDR]**, our main structure is depicted in Figure 1. We bring all the related works from these three main perspectives of cognitive, evaluation (theory and practice), and models together. The cognitive aspect and the measures of compositionality in computational models have very limited research, which is why we rely on the conceptualization provided by Dankers et al. for that part. In the new version, we added the continuity and emerging intelligence aspects to the picture. We connect our claims in the discussion section more clearly to our new version's overviews and evidence in the literature. We hope the changes explained above address this general issue raised by the reviewer.
>
> 1. **[citep vs citet]** We have fixed it according to the context.
> 2. **[related works]** For the theoretical work, we added them in section 5.3 of the revised version as one of the recent papers that established the framework for identifying emergent ability in LLMs. We have also added the evaluation work (Skill-Mix) at the end of Section 3.1 and made connections to it in "Misalignment of Performance and Compositional Learning" in Section 6.

---

> ### Comment · Reviewer_pMmS · 2024-09-13
>
> I appreciate the revisions and believe they have substantially improved the paper. I have changed my Claims And Evidence to Yes to reflect this.

---

### Review · Reviewer_RzbB · 2024-08-28

**Summary Of Contributions:**

The paper surveys compositional learning of AI models, covering the 5 key concepts behind compositionally based on Dankers et al. (2021), important benchmarks for these concepts, popular machine learning approaches studied for compositionally, including classic neural networks, transformers (large language models) and neural-symbolic methods, as well as theoretical findings on these models.
It is motivated to provide "systematic, theoretical and experimental research methodologies" to help "analyze the compositional learning abilities of computational models".

**Audience:**

Yes

**Claims And Evidence:**

Yes

**Requested Changes:**

- `\citep` and `\citet` are used wrongly in the entire draft. Please fix them.
- There are a lot of inconsistent capitalization in the draft. For example, in the last paragraph of section 1, there are "For cognitive aspects", "For Evaluation" and "For Models", where only the first one keeps lowercase.
- Section 2 is heavily dependent on Dankers et al. (2021)
    - Why it is chosen? Are those the only defined or widely accepted metrics of compositionally?
- Figure 1 should be well explained in its caption, e.g. more descriptions are needed to explain the colors, boxes, arrows, etc. Consider converting the second paragraph of section 6 into a compact version as the caption.
- Section 2.1, 2.2 and 2.3: Perhaps works that test those metrics should be listed there.
- Section 2.4: Unlike the end of 2.3, there is no comments on if this is well-tested or not.
- Section 2.5
    - "A well-known example of this is the past-tense debate": What is the debate?
    - "Another example of this task is translating idioms": Can you give an example?
    - Unlike the end of 2.3, there is no comments on if this is well-tested or not.
- Section 2.6: This section should not be at the same level of 2.1-2.5 otherwise it may look like the 6th metric. Perhaps 2.1-2.5 should be put under the same section, considering each of their length is relative short.
- Section 3.3: "PCFG SET Hupkes et al. (2020) is the only benchmark" -> Consider making it softer by "To the best of our knowledge, "
- Table 1: Consider making the 2nd and 4th vertical bars double to make it clear that those are distinct sections (`Name || Text or Multi || 5 metrics`)
- Section 4: The OpenAI citation on GPT-4 has the wrong date, which should be 2023.
- Section 4.1: Perhaps which datasets are used for which method should be listed here.
- Section 4.2: Why some of the tasks here in the second paragraph are not included in Table 1?
- Section 4.3
    - Inconsistent spelling
        - VisProg v.s. Visprog
        - neurosymbolic v.s. neuro-symbolic
    - Missing related work: HOUDINI [1] is an earlier work that combines individual neural networks as sub-modules and learn to solve complex tasks.

[1] Valkov, Lazar, Dipak Chaudhari, Akash Srivastava, Charles Sutton, and Swarat Chaudhuri. "Houdini: Lifelong learning as program synthesis." Advances in neural information processing systems 31 (2018).

**Strengths And Weaknesses:**

# Strengths
- The problem that this survey paper is trying to address is important and having a systematic way to develop compositional AI methods would benefit the community.

# Weaknesses
- The survey is not comprehensive and seems to be opinionated. For example, section 2 is mainly based on a single paper.
- The writing of the paper has a lot of space to improve - see **Requested Changes**.

---

> ### Author Response · Authors · 2024-09-12
>
> Thank you for your valuable suggestions on this paper and for acknowledging its strengths. We have revised the paper based on the reviewers' detailed suggestions and comments. Overall, the revised parts of the paper are highlighted in blue text in the new uploaded PDF.
>
> For the **weaknesses [incomprehensive]**,
>
> We have covered a broad range of literature on compositional learning and structured them in three major aspects (i.e. cognitive, evaluation [theory and practice] and computational models) as depicted in Figure 1. We agree that the compositional learning facets (related to the cognitive aspect) in our old version were mostly based on one work [Dankers et al], this is because Dankers et. al. covers the basic concepts that are widely agreed upon in the later literature and new datasets basing their evaluation splits on this, such as some of our examples in Section 3. However, in our new version, we have revised and expanded that section. Currently, Section 2.1 groups the main tasks of compositionality based on the same Dankers et al and we have added the justification in Section 2.1 introductory paragraph. Additionally, we have added sections 2.3 and 2.4 in the new version where we explain the connection of compositional learning to continuality and emergent intelligence. These two concepts were questioned by the reviewers and we found these concepts most relevant to section 2 which covers basics.
>
> **We have addressed your requested changes in the following ways.**
>
> - **[citep vs citet]** We have changed it according to the context.
>
> - **[capitalization]** We have fixed those. Thanks for the comment.
>
> - **[Section 2]** Currently, Section 2.1 groups the main tasks of compositionality based on the Compositionality Decomposed paper, which to our knowledge, is one of the only formal measures of compositionality proposed that draws from other older existing works. These are becoming widely accepted measures with new datasets basing their evaluation splits on this, such as some of our examples in Section 3. We have highlighted this point now in the Section 2.1 introductory paragraph. We have added sections 2.3 and 2.4 in the new version, where we explain the connection of compositional learning to continuality and emergent intelligence. These two aspects were questioned by the reviewers.
>
> - **[Figure 1 Caption]** We have added a more descriptive caption based on your suggestion.
>
> - **[Section 2.1- 2.3]** We cover datasets that use these metrics in Section 3. It may be redundant to include them in Section 2 as well. We are open to repeating them in the final version if the reviewer still thinks that makes the paper clearer.
>
> - **[Section 2.4]** We have added a line at the end of the 3rd paragraph of Section 2.1.
>
> - **[Section 2.5]**
>     - **[debate]** We have clarified this in the text with blue color in the 5th paragraph of Section 2.1.
>     - **[idioms]** We have clarified this in the text with blue color in the 5th paragraph of Section 2.1.
>     - **[well-tested]** We have added a line at the end of the 4th paragraph of Section 2.1.
>
> - **[Section 2.6]** We have modified our section organization to be easier to understand and grouped our original Section 2.1-2.5 under the same section as suggested.
>
> - **[Section 3.3]** We have changed the language. Thanks for the comment.
>
> - **[Table 1]** Thank you for the suggestion, we have applied it.
>
> - **[GPT citation]** We are using the updated version published in 2024, so the citation should be okay.
>
> - **[Section 4.1]** Some papers in the section have tasks that need to be included in our dataset list. We clarified this at the end of our introductory paragraph for Section 4. For the rest, we made sure the tasks and datasets were mentioned.
>
> - **[Section 4.2]** Due to a lack of space, we picked examples instead of an exhaustive list, and many of the related papers are cited in the paper's content. We found a subset more practical and easier to present in a table.
>
> - **[Section 4.3]**
>     - **[spelling]** We have fixed those. Thanks for the comment.
>     - **[related work]** We have added a new paragraph including this work under Compositional Neural Architectures paragraph in Section 4.3. We have also extended this reference and made a connection to continual learning in Section 2.3.

---

### Decision · Action_Editor_8F89 · 2024-10-17

**Recommendation:** Accept as is

**Comment:**

The authors have
- rescoped the work to make it clear the focus is on language based benchmarks/evaluation
- updated the citation formatting
- expanded discussion on measures
- improved structure
- included discussion of continual learning and "emergent" intelligence
- Expanded discussion more generally, including on less explored facets
- Addition of MIT-States, etc

These address the main concerns for the current manuscript.

**Audience:**

Those interested in compositional representations within the language space, potentially informed by experiments and theories from linguistics and cogsci but applied to the current (V)LM setting.

**Claims And Evidence:**

This is a survey which aims to outline areas with insufficient research regarding compositionality and generalization, targeting questions of evaluation and theory.  A categorization is provided as well as discussion of specific datasets.  Relevant aspects of these datasets are highlighted.

I will note that while length generalization comes up a number of times in the work, there is very little discussion about the work in this space on the model/architecture side.  I don't think this is necessary but the section is rather short and there are a number of works (specifically in the domain of transformers) that do discuss/explore such issues. I will leave a note with some references for the authors for future reference.